

# BERT4Cache: a bidirectional encoder representations for data prefetching in cache

Jing Shang, Zhihui Wu, Zhiwen Xiao, Yifei Zhang and Jibin Wang

China Mobile Information Technology Center, Beijing, China

## ABSTRACT

Cache plays a crucial role in improving system response time, alleviating server pressure, and achieving load balancing in various aspects of modern information systems. The data prefetch and cache replacement algorithms are significant factors influencing caching performance. Due to the inability to learn user interests and preferences accurately, existing rule-based and data mining caching algorithms fail to capture the unique features of the user access behavior sequence, resulting in low cache hit rates. In this article, we introduce BERT4Cache, an end-to-end bidirectional Transformer model with attention for data prefetch in cache. BERT4Cache enhances cache hit rates and ultimately improves cache performance by predicting the user's imminent future requested objects and prefetching them into the cache. In our thorough experiments, we show that BERT4Cache achieves superior results in hit rates and other metrics compared to generic reactive and advanced proactive caching strategies.

## INTRODUCTION

In the past few years, Internet services have undergone exponential growth in terms of user numbers, applications, and data traffic. Its service scope has expanded from multimedia streaming to online gaming (*Liu et al., 2018*). As the existing network architecture faces challenges in efficiently handling massive data transfers with minimal latency requirements (*Shafi et al., 2017*), as a technique which can enhance system response rates and reduce redundant network traffic, caching plays an increasingly vital role in the current network architecture.

Caching is a technique for temporarily storing data. It operates by storing copies of previously retrieved or computed data in high-speed storage media, providing direct access to the required data objects during data retrieval without waiting for the time it takes to retrieve them from slower storage medium, which reduces the accessing frequency, enhancing system response speed and availability. Caching has been used in many scenarios. In computer systems, CPUs often utilize multilayer high-speed cache to store instructions and data, to accelerate data processing. Server caching is employed to store previously retrieved network resources, reducing access latency to remote servers. Web services, content delivery networks (CDNs), and technologies like RL-Cache (*Kirilin et al., 2019*) use

Corresponding author
Zhiwen Xiao,
xiaozhiwen15@mails.ucas.ac.cn

caching to enhance loading and response speeds. In file systems, caching is employed to store recently read or written file data, minimizing disk access, as seen in the heterogeneous memory cache framework (FLAC) (*Zhang et al., 2020*).

The architecture of caching varies based on specific application scenarios. For instance, server caching may adopt centralized, distributed, or multi-level cache architectures. In many large-scale Internet storage systems, memory-based caching systems like Memcached (opensource, 2024), Redis (opensource, 2024), or local caching libraries such as Ehcache (M LGSSKGT, 2023) are deployed to ensure service-level objectives (SLOs).

Despite the various applications of caching, they all leverage a common advantage of caching: storing copies of data that might be accessed in the future in high-speed media. Therefore, the effectiveness and efficiency of caching depends on whether the caching data will actually be accessed in the future. In other words, the efficacy of caching directly relies on its hit rate, and the hit rate is constrained by the accuracy of cache algorithms in predicting the popularity of content in the imminent future. Consequently, modeling the popularity of content in the imminent future, predicting it by analyzing historical sequence data, and designing cache algorithms based on this prediction have become key research areas in designing efficient caching systems.

Given the vast number of objects that may be requested in the future, one of the primary challenges faced by caching algorithms is effectively determining which objects should be stored in the cache. Based on the approach used for cache decision-making, caching algorithms can be categorized into four main types: Rule-based caching, Data mining-based caching, Classical machine learning-based caching, and Deep learning-based caching. Rule-based caching involves low computational overhead and fast response times but cannot learn user interests and preferences. Data mining-based caching and classical machine learning-based caching, while capable of learning user access patterns to some extent, are limited by the algorithms themselves in finding optimal solutions and may not delve deeply into pattern mining. Deep learning-based caching, on the other hand, achieves a deeper understanding of user behavior and patterns through deep learning techniques. It possesses stronger learning capabilities and adaptability, allowing for more accurate predictions of objects that might be requested in the future. It excels in handling large-scale data, automatically extracting and comprehending complex patterns in the data. Despite its higher computational complexity, with the widespread increase in server computing power and the gradual adoption of edge computing architectures (*Cao et al., 2020*), deep learning-based caching has shown vast application prospects. It has garnered attention from researchers for its ability to handle evolving user demands and access patterns effectively.

According to the scope of information on which decisions rely, caching algorithms can be broadly categorized into two types: reactive caching and proactive caching. Reactive caching makes decisions based on real-time, current information, and request context, usually at the time of request arrival. When a new request arrives, a reactive caching algorithm considers the current cache state and the request content to determine whether to place the object in the cache and whether to replace existing objects in the cache. Rule-based Least Recently Used (LRU), Least Frequently Used (LFU), and their variants are

representative of reactive caching. These algorithms make cache decisions solely based on the recently observed object access patterns. However, a key drawback of reactive caching is that it can only perceive the popularity of an object after it has been requested, making it challenging to capture dynamic changes in popularity in a timely manner. Therefore, recent research has gradually shifted focus towards proactive caching.

Proactive caching is based on historical request information, predicting and pre-storing data objects in the cache before actual requests arrive. It involves analyzing historical access patterns, content popularity trends, and other information to determine which content should be preloaded into the cache. The proactive approach often gains global knowledge of object access patterns and predicts imminent future requested objects based on the recent object access sequences during the inference process. Traditional proactive caching methods commonly utilize data mining and shallower machine learning techniques, such as dependency graph-based methods, Markov model-based methods, recurrent neural network (RNN) methods, and their variants. In recent years, some emerging deep learning-based proactive caching methods have drawn researchers' attention, including methods based on attention mechanisms.

The effectiveness of proactive prefetching largely depends on the accuracy of predicting the popularity of content in the imminent future. Existing methods model the entire cache access history sequence directly without partitioning the cache access history sequence into multiple independent single-user sub-sequences, which prevents the model from distinguishing between the commonalities in access patterns across users and the unique characteristics of each individual user. Such lack of differentiation may lead to suboptimal learning of user behavior, as the model would be forced to generalize across all users without considering the specific nuances of each user's access patterns. Besides, since user access requests do not occur at uniform time intervals, the overlay of multiple user request sequences can confound the pattern information of user requests. This confounding effect leads to the instability of content popularity from the perspective of the entire system. Additionally, user's historical interactions in real-world applications may not follow a rigid order assumption (*Wang et al., 2018*). Active caching methods based on self-regressive structures like RNN, LSTM, *etc.*, employ unidirectional models that limit the hidden representation capacity of items in the historical sequence. This limitation is insufficient for learning the optimal representation of user behavior sequences (*Hu et al., 2017*), as the bidirectional Transformer architecture, similar to BERT, is capable of simultaneously considering the information contained in the tokens preceding and following each token during the encoding process. This enables the model to better comprehend the patterns within the access sequence, thereby making more accurate predictions. Therefore, it is necessary to partition the access sequences into user sessions and introduce bidirectional context representations in active caching algorithms to address these issues.

To tackle these problems, we introduce BERT4Cache, which utilizes a bidirectional Transformer network with attention mechanisms to learn user access patterns. It predicts the most likely objects to be accessed in the imminent future based on the user's historical access and guides cache prefetching to improve cache hit rates. The contributions of our article are as follows:

1) Proposed BERT4Cache, an end-to-end bidirectional Transformer model with attention mechanisms, to explore users' access patterns, predict objects users are likely to request in the near future, and prefetch them into the cache.

2) Introduced forward consistency to enhance user access sequences during data modeling. Combined with a mask-based sequence prediction task, the model is trained to learn users' data access patterns.

3) Built a simulated caching system and demonstrated, using real-world data, that BERT4Cache achieves advanced performance compared to generic reactive and advanced proactive caching strategies.

## RELATED WORKS

### Classical caching updating

A significant amount of research has been conducted on caching algorithms, with examples including reactive caching algorithms like LRU, LFU, and their variants, which are commonly used in processor memory caches and web caches. Some studies have proposed proactive caching algorithms based on rules and data mining, such as graph-based caching, Markov model-based caching, and cost function-based caching. Some researchers employed a dependency graph method to predict the popularity of requested objects. However, the dependency graph only examines dependency relationships between two objects, leading to lower prediction accuracy. To address issues such as low modeling hierarchy for low-order Markov models and low coverage for high-order Markov models, the full k-order Markov model was introduced. However, the full k-order Markov model exacerbates complexity problems as the states of all Markov models are additive. *Khan et al. (2023)* proposed SHADE, which dynamically updates the importance scores of all samples during training to make informed caching decisions, thereby improving the cache hit rate for deep learning training tasks. *Einziger, Himelbrand & Waisbard (2023)* introduced an adaptive access time-aware caching strategy, using changes in access time as an additional signal for the caching algorithm, thereby improving the average access time of web cache. *Li et al. (2022)* mined frequent block access patterns and compared the requests in the current time window to direct prefetching data into the cache of SSDs. However, cost function-based and data mining-based algorithms introduce too many manually observed patterns, exhibiting weak robustness across different contexts and struggling to effectively learn user interests and preferences.

### Machine learning-based caching updating

As a powerful tool, machine learning algorithms can learn the popularity of data objects, the mobility of user preferences, content characteristics of data objects, and consequently cache the most popular files while removing the least likely to be accessed files to enhance hit rates. The machine learning algorithm used in this process can be a self-regressive method, a reinforcement learning-based approach, and so on. *Ale et al. (2019)* proposed an online proactive caching scheme based on a bidirectional recurrent neural network (BRNN) model to predict temporal content requests and update the cache accordingly.

*Narayanan et al. (2018)* improved cache hit rates by applying the DEEPCACHE content caching framework, based on deep LSTM encoder-decoder, to predict object popularity and proactively cache the most needed items in the future. *Wei et al. (2018)* used reinforcement learning to learn the optimal caching strategy through interaction with the network environment. *Rodriguez et al. (2021)* modeled dynamically changing workloads as a composition of four workload primitive types. They employed reinforcement learning to select multiple caching strategies such as ARC, LIRS, adjusting their weights to achieve adaptive cache updates. *Yang et al. (2023)* clustered similar objects into groups and perform cache prefetching and replacement operations based on the features of these groups. *Navarro et al. (2017)* run a set of applications on a processor simulator, obtained their instruction features, and trained a model based on this to predict the optimal cache reconfiguration for any given application. *Huang et al. (2016)* introduced a "one-time-access criteria" applied to the cache space. They propose a policy to exclude unnecessary SSD cache writes and design a prediction-based classifier to support the policy. *Zarif et al. (2020)* employed an RNN-based LFU and proposed a proactive caching framework independent of the database system. This framework predicts the time cost and frequency of queries to guide the caching process. However, these methods do not distinguish between users, and complex access sequences may confound pattern information of user requests, affecting cache hit rates. Constrained by their inherent characteristics, such algorithms struggle to find the optimal solution for cache updates and may not delve deep enough into pattern mining. Additionally, these methods face the challenge of low inference efficiency due to difficulties in parallel computing.

## Attention mechanism and proactive prefetching

The attention mechanism dynamically adjusts the weights assigned to different parts of input data based on their distinct features. This enables selective focus during data processing, making the model more effective in exploring long-range dependencies and complex patterns within sequences. With models like Transformer (*Vaswani et al., 2017*), BERT (*Devlin et al., 2019*), *etc.*, built on multi-head self-attention achieving leading results in sequence data modeling such as machine translation and text classification, some researchers are exploring the use of attention mechanisms to enhance the hit rate and interpretability of recommendation systems. *Li et al. (2017)* proposed the SASRec model, based on a Transformer decoder, to learn user browsing behavior, achieving state-of-the-art results across multiple public datasets. *Sun et al. (2019)*, through a cloze task, modeled user behavior sequences using a bidirectional self-attention network, demonstrating significant potential in sequence recommendation. Both sequence recommendation and proactive prefetching require modeling user data access patterns to predict future user behavior before actual requests. Therefore, the combination of attention mechanisms and proactive prefetching has become a popular topic in recent years.

## THE PROPOSED BERT4CACHE METHOD

In this article, our objective is to leverage sequentialized data for the purpose of characterizing user request sequences through the application of BERT with integrated

forward consistency. Subsequently, we employ this characterization to construct a predictor based on user request sequences, thereby guiding the prefetching process of cache prefetching.

To this end, the proposed methodology comprises two primary stages: (1) Data modeling, involving the transformation of continuous, unsegmented raw log data into forward consistency pairs (FCP) and sequence-prefetch sets (SPS). The detailed procedure will be elucidated in "Data Modeling"; (2) Training, which entails the utilization of masked data sequence (MDS) and forward consistency (FC) tasks to train a BERT network that is particularly attuned to recent request history. The specific steps will be expounded upon in "Training BERT4Cache".

## Problem formulation

In the context of sequentialized data $L$, let $U = \{u_1, u_2, ..., u_{|U|}\}$ denote the set of users, and $O = \{o_1, o_2, ..., o_{|O|}\}$ represent the set of objects to be prefetched. For a given user $u \in U$, their request sequence set $\hat{S}_u = \{s_1, ..., s_N\}$ is obtained by slicing the sequences with a maximum time interval $g$, where $N$ is the number of sequences after slicing. For each sequence slice $s \in \hat{S}_u$, $s = [o_1, ..., o_t]$, where $o_t$ is the object last requested in the slice according to the time order. Given the data $L$, cache prefetching aims to capture the latest sequence slice $s_{now}^{u_i}$ for each user $u_i$ and model all possible items for the next request by that user, as shown in Eq. (1).

$$p(o_{t+1}^{u_i} = o | s_{now}^{u_i}). \tag{1}$$

## BERT4Cache architecture

In this section, we present an innovative framework, Bidirectional Encoder Representations from Transformers for cache prefetching (BERT4Cache).

As depicted in Fig. 1, BERT4Cache comprises five essential components: (1) Sequentialized data processor, responsible for transforming sequentialized data into request object sequences organized by user sessions; (2) forward consistency pair generator, leveraging a sliding window approach to construct FCPs based on the sequences obtained in step (1); (3) Sequence-Prefetch Set Generator, tasked with appending a special token [mask] to the end of each request object sequence and simultaneously replacing random $p\%$ of tokens in the sequence with the special token [mask]; (4) BERT4Cache Network, trained on the data generated in steps (2) and (3) using the Forward Consistency Task and Cloze Task to enable the model to learn complex patterns in user's object requests. Specifically, the main network architecture of BERT4Cache comprises multiple layers of bidirectional Transformer blocks. Each block is constructed with multi-head self-attention layers, capturing implicit relationships among encoded representation tokens. In this study, we employed 12 Transformer blocks, each incorporating 12 attention heads. The dimensionality of each input was set to $D = 768$, with a total of $L = 512$ input tokens; (5) Decoding Network, responsible for decoding the context-encoded results from the BERT4Cache Network into actual categories of objects that can be prefetched in the cache

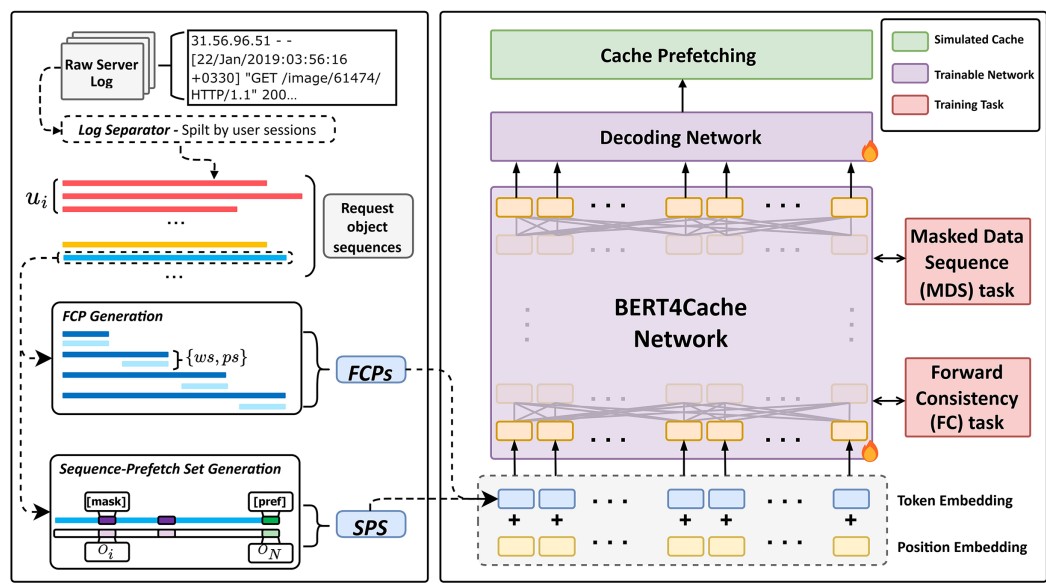

**Figure 1** **The overall architecture of BERT4Cache consists of two main components: Data modeling, which transforms raw logs into FCP and SPS; Training, which utilizes MDS and FC tasks to train the BERT4Cache network.** BERT4Cache comprises five essential components: (1) Sequentialized data Processor, responsible for transforming sequentialized data into request object sequences organized by user sessions; (2) Forward consistency pair generator, leveraging a sliding window approach to construct forward consistency pairs (FCPs) based on the sequences obtained in step (1); (3) Sequence-Prefetch Set Generator, tasked with appending a special token [mask] to the end of each request object sequence and simultaneously replacing random $p\%$ of tokens in the sequence with the special token [mask]; (4) BERT4Cache Network, trained on the data generated in steps (2) and (3) using the Forward Consistency Task and Cloze Task to enable the model to learn complex patterns in user's object requests. Specifically, the main network architecture of BERT4Cache comprises multiple layers of bidirectional Transformer blocks. Each block is constructed with multi-head self-attention layers, capturing implicit relationships among encoded representation tokens. In this study, we employed 12 Transformer blocks, each incorporating 12 attention heads. The dimensionality of each input was set to $D = 768$, with a total of $L = 512$ input tokens; (5) Decoding Network, responsible for decoding the context-encoded results from the BERT4Cache Network into actual categories of objects that can be prefetched in the cache system. In this study, we employ a two-layer feed-forward network with GELU activation in between to produce a $|O|$-dimensional output distribution.

system. In this study, we employ a two-layer feed-forward network with GELU activation in between to produce a $|O|$-dimensional output distribution.

## Transformer layer

As described in "BERT4Cache Architecture", the BERT4Cache Network is constructed by stacking multiple bidirectional transformer blocks. Each bidirectional transformer block can be divided into two components: a multi-head self-attention sub-layer and a position-wise feed-forward network.

### Multi-head self-attention

The self-attention mechanism has been demonstrated as an effective method of modeling sequence information. Multi-head self-attention, building upon the self-attention mechanism, introduces multiple independent attention heads to capture diverse information scales and relationships, thereby significantly enhancing the model's

representational capacity for sequence information. By incorporating multiple attention heads, the model can concurrently learn different attention weights, enabling the effective processing of both local and global information in the sequence.

Specifically, given an input sequence $X$ of length, the BERT4Cache Network, composed of multiple stacked transformer blocks, iteratively computes the hidden representation $h_i^l \in R^d$ for each position $i$ and each layer $l$, where $d$ denotes the dimensionality of the hidden layer. The hidden representations calculated by all transformer blocks at the same layer are concatenated to form the hidden representation matrix $H^l \in R^{n \times d}$ used for attention calculations across all positions in the same layer.

Multi-head self-attention employs learnable linear projections to project the hidden representation matrix $H^l$ into $h_c$ subspaces, where $h_c$ is the number of attention heads. Subsequently, $h_c$ attention functions are applied in parallel to generate output representations. The output of each attention head is concatenated, and a linear transformation is then applied to map the output of multi-head self-attention to the desired dimensionality, forming the final output of multi-head self-attention, as shown in Eq. (2).

$$MultiHead(H^l) = Concat(head_1, head_2, \ldots, head_{h_c})W^O$$
$$head_i = Attention(H^l W_i^Q, H^l W_i^K, H^l W_i^V). \tag{2}$$

Here, $W_i^Q \in R^{d \times d/h_c}$, $W_i^K \in R^{d \times d/h_c}$, $W_i^V \in R^{d \times d/h_c}$ are the learnable weight matrices for the $i$-th attention head, and is the output weight matrix. Scaled Dot-Product Attention is used as the attention function, as shown in Eq. (3).

$$Attention(Q, K, V) = softmax\left(\frac{QK^T}{\sqrt{d/h_c}}\right)V. \tag{3}$$

During training, due to the potential for gradient explosion or vanishing caused by parameter updates in neural networks, the softmax computation introduces a temperature parameter to adjust the sharpness of the probability distribution output by the softmax function, as shown in.

$$softmax(x) = \frac{e^{x_i/\tau}}{\sum_j e^{x_j/\tau}}. \tag{4}$$

Here, $x$ is the input vector, and $\tau$ is the temperature parameter. A higher temperature $\tau$ results in a smoother output distribution from the softmax function, while a lower temperature sharpens the distribution.

### Position-wise feed-forward network

To enhance the model's capability in capturing information at each position and improve its representational power over sequences, a position-wise feed-forward network is introduced within the transformer block for non-linear transformation of the hidden representation $h_i^l$ at each position in the sequence, as shown in Eq. (5).

$$FFN(h_i^l) = GELU(h_i^l W_1 + b_1)W_2 + b_2. \tag{5}$$

Here, $W_1 \in R^{d \times 4d}$, $W_2 \in R^{4d \times 4d}$, $b_1 \in R^{4d}$, and $b_2 \in R^d$ are all learnable parameters. Following the approach of BERT, we utilize the GELU activation function.

## Data modeling

In this section, we introduce the methodology employed for effective data modeling from log data.

### *Sequence partition*

In the research on cache prefetching using pre-trained models, especially in methods based on log data, a crucial step involves the partitioning of logs. In this step, we employ a strategy of partitioning based on user sessions, dividing the raw logs into sets of user sessions by setting a time difference threshold.

As illustrated in Fig. 1, firstly, we consider the user as the fundamental unit and partition the raw logs based on users. This step aims to capture the behavioral patterns of each user during the request process, segregating them from the patterns of other users. This demarcation prevents the model from failing to learn representations of user request processes due to the mixing of user patterns.

Specifically, the log data $L$ can extract independent user sets $U = \{u_1, u_2, ..., u_{|U|}\}$, where $u_i$ represents the user ID appearing as an attribute in the log data entries. Based on the user set, the objects in the log can be divided into a collection $C = \{S_{u_1}, ..., S_{u_i}, ..., S_{u_{|U|}}\}$ consisting of $|U|$ access sequences $S_{u_i}$ arranged in chronological order. For each $o_i \in S_{u_i}$, it satisfies $\forall o_i \in S_{u_i}, req(o_i) = u_i$, where $req(o_i) = u_i$ indicates that the object $o_i$ has been requested by user $u_i$, which means this request is recorded in the log data $L$.

Subsequently, we use a time difference threshold $g$ as a criterion to further partition the request sequence $S_{u_i}$ of the same user into a collection of sub-sequences $\hat{S}_{u_i}$. The setting of the time difference threshold is based on the temporal characteristics of user requests in the log data used. By sessionizing the request sequence of individual users, we can better capture changes in user behavior over different time periods, enhance multi-scale information of user requests, and improve the modeling effectiveness of request temporal dynamics.

Specifically, for each object in the request sequence $S_u$ of the user $u \in U$, if the time difference between it and the preceding object is less than or equal to the threshold $g$, it is added to the same sub-sequence $s_j$. When the first object with a time difference greater than $g$ appears, a new sub-sequence begins. Through iterating this process, we obtain a series of sub-sequences $\hat{S}_u = \{s_1, ..., s_N\}$ that satisfy the time difference threshold condition. For each sequence slice $s \in \hat{S}_u$, $s = [o_1, ..., o_t]$, where $o_t$ is the object last requested in the slice according to the time order.

### Forward consistency pair generator

Based on the sub-sequence collection $\hat{S}_u = \{s_1, ..., s_N\}$ generated in "Multi-Head Self-Attention", which satisfies the time difference threshold condition, we use the forward consistency pair generator to construct FCP, adapting to the FC task during network training, which will be introduced in "Data Modeling". FCP enables us to improve the directional focus of the network during the training of a BERT-like Bidirectional Encoder Representations Network, enhancing its adaptability to cache prefetching tasks.

As illustrated in Fig. 1, specifically, for each sequence $s_i \in \hat{S}_u$, we employ a sliding window for sampling. Starting from the $i$-th element of, we extract elements including and following, forming a windowed sub-sequence $ws = \{o_j, ..., o_{j+w}\}$, and construct a truncated sub-sequence $ps = \{o_1, ..., o_{j+w}\}$, defining an FCP as $P = \{ws, ps\}$. We update the value of $j$ to $j + p$ and repeat this process until $j + w \geq |\hat{S}_u|$. Thus, $s_i$ generates $\lceil (l - w)/p \rceil$ + FCPs. The purpose of this step is to align with cache prefetching scenarios. In practice, the length of the input object sequence that determines the cache prefetch results is often limited. Therefore, the forward consistency pair generator applies the same to each sequence in $\hat{S}_u$ to ensure that the training scenario closely matches the real-world application scenario. This ensures the model's encoding capability for variable-length inputs, thereby improving its adaptability to cache replacement scenarios.

### Sequence-prefetch set generator

We introduce the Sequence-Prefetch Set (SPS) generation process. This method is particularly beneficial in scenarios where the length of the input object sequence significantly impacts cache prefetch results. The Sequence-Prefetch Set is designed to capture the temporal correlations among recently accessed objects, allowing the model to effectively prefetch objects that are likely to be requested in the near future.

Subsequently, similar to BERT4Cache, we also randomly replace $\lambda\%$ positions of the sequence model with a special token [mask] and require the model to complete them to learn complex patterns in user object requests. The difference from BERT4Cache is that during the training phase, we have already replaced the last token of each object token sequence with the special token [pref]. By explicitly requiring the model to differentiate between "summarize" and "predict" behaviors, we prevent mismatches between training and the final cache prefetch scenario, as the Cloze objective aims to predict the current masked items, while the Predict objective aims to predict the future. Additionally, FCPs generated according to the method described in "Position-wise Feed-Forward Network" will also be part of the training data to enhance the model's predictive capabilities for fixed-length token request sequences that may not include semantics indicating the start of a pattern.

In FCPs, the $\eta\%$ positions of $ws$ will also be replaced with the special token [mask], and its last token will also be replaced with the special token [pref]. Setting different mask ratios is aimed at encouraging the model to learn complex patterns by predicting the masked tokens within the fixed-length sequence.

The sequence with added special tokens, along with the original token values at the positions replaced with special tokens [mask] and [pref], is combined into SPS, denoted as $\{S_c, S_p\}$.

## Training BERT4Cache

We propose two training tasks: MDS task and FC task, and train the BERT4Cache network accordingly. The former captures the context relationship between request objects by predicting [mask] and [pref] tokens. The latter enhances the directional attention of the network for better adaptation to cache prefetch tasks by replacing the end of FCP with [pref] tokens and minimizing the similarity of their [pref] token encoding results. Truncation and padding are utilized to maintain a consistent sequence length for input into BERT4Cache. The detailed process is shown in the middle of Fig. 1.

**MDS task.** BERT4Cache can learn contextualized representations without the need for methods like Byte Pair Encoding (BPE) to segment input sequences. Each token input into BERT4Cache is an independent unit containing complete semantic meaning. This is determined by the actual task scenario of cache prefetching, where the set of objects used for prefetching is finite and well-defined, and the prefetching target should correspond entirely to an object in the file system.

Specifically, after the Sequence-Prefetch Set Generate step, the original log data is transformed into $SPS = \{S_c, S_p\}$. Before encoding with BERT4Cache, the [cls] special token is added at the beginning of the $S_c$ sequence, and $(L - (|S_c + 1|))$ [pad] special tokens are added at the end to complete the input sequence for the model. This allows the model to differentiate between special tokens without actual semantics and object tokens containing actual semantics, enabling the model to learn the encoding of sequences shorter than the model's maximum input sequence length.

During training, for each sequence in $S_c$, BERT4Cache encodes each token by considering the overall semantic context. Specifically, assuming the $j$-th position in $S_c$ is a [mask] token or [pref] token, BERT4Cache encodes it by considering the overall semantic context, producing an embedding vector $E_j \in R^D$. The Decoding Network decodes $E_j$, yielding the corresponding probability $U_j$. For the $j$-th position, where $S_p[j]$ is the original value, the loss function for this part is given by Eq. (6).

$$L_{MDS} = -\sum_{j=1}^{k} one-hot(S_p[k]) \log(U_j). \tag{6}$$

Here, $one-hot(S_p[j])$ converts $S_p[j]$ into a one-hot encoded vector, where $j$ iterates over the positions of [mask] tokens and [pref] tokens. For $U_j$, the formula is given by Eq. (7).

$$U_j = softmax(W_2 \dot{GELU}(W_1 E_j + b_1) + b_2). \tag{7}$$

Here, $E_j \in R^D$ is the embedding vector generated by BERT4Cache for the input token, considering the context. $W_1$ and $W_2$ are the weight matrices of two feedforward network layers, respectively. $b_1$ and $b_2$ are the corresponding biases.

**FC task.** In cache prefetch tasks, modeling the future behavior of request sequences and capturing directional information are crucial for improving prediction accuracy. Based on the previously described FCP generation process, we design the FC task to enhance the directional attention of the network for better adaptation to cache prefetch tasks. By adding special tokens to the FCP sequence and using the corresponding positions in BERT4Cache embeddings, we calculate the FC Loss. The computation of FC Loss does not require the involvement of the decoding network and is optimized solely using the cosine similarity of embedding results.

Specifically, for the FCP $P_i = \{ws_i, ps_i\}$ generated on log data, we add a special token [cls] at the beginning of the $ws_i$ sequence, append a special token [pref] at the end, and then add $(L - (|ws_i| + 1))$ special tokens [pad], resulting in the sequence $\widehat{ws_i}$. The index of the special token [pref] is denoted as $i_w$. The same procedure is applied to $ps$, yielding the sequence $\widehat{ps_i}$ with the index of the special token [pref] denoted as $i_p$. The sequences $\widehat{ws}$ and $\widehat{ps}$ from the same FCP $P_i$ are embedded using BERT4Cache, and the cosine similarity of the embedding results at positions $i_w$ and $i_p$ is computed as FC Loss, as shown in Eq. (8).

$$L_{FC} = 1 - \cos(E_{i_w}, E_{i_p}). \tag{8}$$

Our aim in designing this is to optimize the directional focus of the BERT4Cache network by minimizing the similarity in FC Loss. This process does not involve the decoding network. $E$ in Eq. (8) is defined in Eq. (9).

$$E_{i_w} = BERT4Cache(\widehat{ws_i})[i_w]$$
$$E_{i_p} = BERT4Cache(\widehat{ps_i})[i_p]. \tag{9}$$

In this context, $BERT4Cache(\widehat{ws_i})$ represents the embedding of the sequence $\widehat{ws_i}$ using BERT4Cache, and $[i_w]$ denotes the extraction of the embedding result at a specific position.

Our design aims to emphasize the network's embedding of the [pref] token at the end of the request sequence in FC Loss, contributing to improved modeling of the future behavior of request sequences. This emphasizes the network's attention to the directional information in request sequences, enhancing its adaptability in cache prefetch tasks.

Overall, the final pre-training objective is the sum of the above two losses, which is defined in Eq. (10).

$$L = \alpha L_{MDS} + \beta L_{FC}. \tag{10}$$

# EXPERIMENTS

This section introduces the implementation method of the BERT4Cache model in the caching system, as well as the optimization effect of the caching system achieved using BERT4Cache.

## Settings and dataset

We implemented a simulated caching system. During testing, simulated requests from different users are sent to the simulated caching system, and the cache hit rate of the

simulated caching system is then calculated. This section describes the implementation of the simulated caching system and the method used to construct simulated user requests.

The simulated caching system consists of four parts: the BERT4Cache communication module, the cache table module, the backend storage communication module, and the cache replacement algorithm module. Upon receiving a user request, the simulated caching system first queries the cache table to check if the requested object is in the cache. If the requested object is not in the cache, it needs to be loaded from the backend storage. During loading, if the cache system does not have enough space, data objects currently in the cache need to be deleted based on the cache replacement algorithm. If the requested data object is in the cache, the simulated caching system calls the BERT4Cache model, inputs the historical access data object sequence of the requesting user into the BERT4Cache, and lets the BERT4Cache model predict the data object the user will access in the immediate future, prefetching the result returned by BERT4Cache into the cache.

The cache replacement algorithm module may use various algorithms. BERT4Cache has different performance improvement effects on different cache replacement algorithms. We evaluated four different replacement algorithms and two cache prefetching algorithms in "Performance Evaluation": LRU, LFU, random replacement, FIFO, ACORN (*Wang et al., 2024*) and ANN-based proactive caching strategies (*Lutfianto, Rizal & Negara, 2023*).

Because LFU replaces data objects based on the number of accesses, and the access count of the data objects preloaded into the cache but not yet accessed is 0, they are easily replaced directly by LFU, resulting in preload failure. We improved the original LFU. The improved LFU prioritizes selecting the data object to be replaced from the data objects loaded before 10,240 user requests. If a data object to be replaced cannot be determined from these data objects, it is then selected from the remaining data objects.

We used the MovieLens dataset (ML-1m) (*Harper & Konstan, 2015*) to construct the access sequences of the simulated caching system. Although originally designed for recommender system research, the dataset's user-movie interactions, such as watching and rating, are analogous to access request sequences in caching scenarios. For each user in the ML-1m dataset, we extracted their access records, sorted these access records based on access time, and obtained the access sequence for each user. The obtained access sequences are sent to the cache simulation system, and the cache hit rate of the cache simulation system is calculated. In "Performance Evaluation", we first evaluated the performance under single-user access scenarios (A, B, C), and then evaluated the performance under multi-user access scenarios (D).

## Performance evaluation

We configured different parameters for the cache simulation system to simulate different scenarios and observed the cache hit rate under different scenarios to validate the performance improvement effect of BERT4Cache on the caching system.

The cache hit rate is calculated as the ratio of cache hits to total accesses, *i.e.*, $R = N_h/N$ where $N_h$ is the number of cache hits and N is the total number of accesses. To compare the cache hit rates between scenarios with and without using BERT4Cache, we calculated the improvement ratio of BERT4Cache on the caching system performance using $\frac{r'-r}{r}$, where $r'$

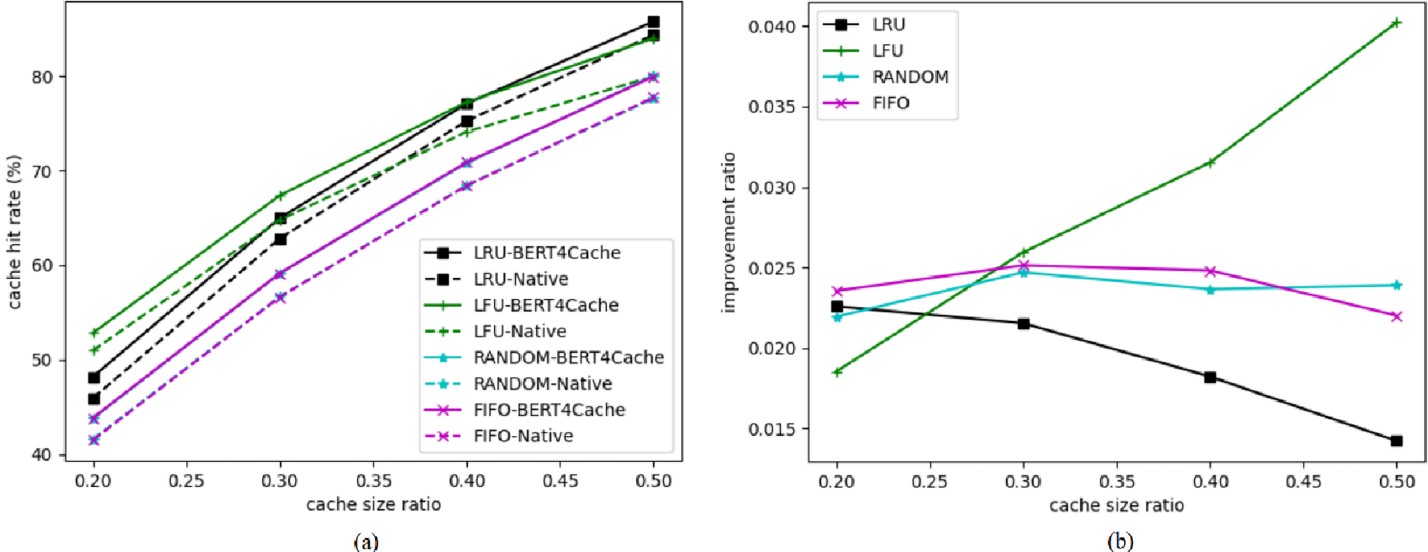

**Figure 2 Cache hit rates for different cache replacement algorithms.** The horizontal axis represents the cache ratio, _i.e._, the ratio of the cache size to the size of the backend storage, and the vertical axis represents the cache hit rate. The curves ending with 'BERT4Cache' denote the cache hit rates corresponding to the use of BERT4Cache, while other curves denote the cache hit rates when BERT4Cache is not used. It can be observed that as the cache ratio increases, the cache hit rate gradually increases.

is the cache hit rate when using BERT4Cache, and $r$ is the cache hit rate when not using BERT4Cache. In the following subsections, we present the performance evaluation of BERT4Cache under different cache sizes, access frequencies, prefetch ratios, and multi-user scenarios. Additionally, we provide in-depth analysis and discussions to explore the underlying reasons for the observed results.

### Performance comparison of different cache sizes

The cache hit ratio varies with the cache size of the cache system. Generally, a larger cache size results in a higher cache hit rate. We configured the simulated caching system with different cache sizes and input the same user access sequence. Under different cache sizes, we selected different cache replacement algorithms and compared the cache hit rates when using BERT4Cache and when not using BERT4Cache.

As shown in Fig. 2, the horizontal axis represents the cache ratio, _i.e._, the ratio of the cache size to the size of the backend storage, and the vertical axis represents the cache hit rate. The curves ending with 'BERT4Cache' denote the cache hit rates corresponding to the use of BERT4Cache, while other curves denote the cache hit rates when BERT4Cache is not used. It can be observed that as the cache ratio increases, the cache hit rate gradually increases. The improvement ratio of the cache hit rate decreases with the increase of the cache ratio, indicating that BERT4Cache has a consistent improvement effect on the cache hit rate. This phenomenon can be explained by the fact that with the expansion of cache space, more objects can be cached, thus improving the performance of baseline algorithms as well. Consequently, the advantage of BERT4Cache becomes relatively less pronounced.

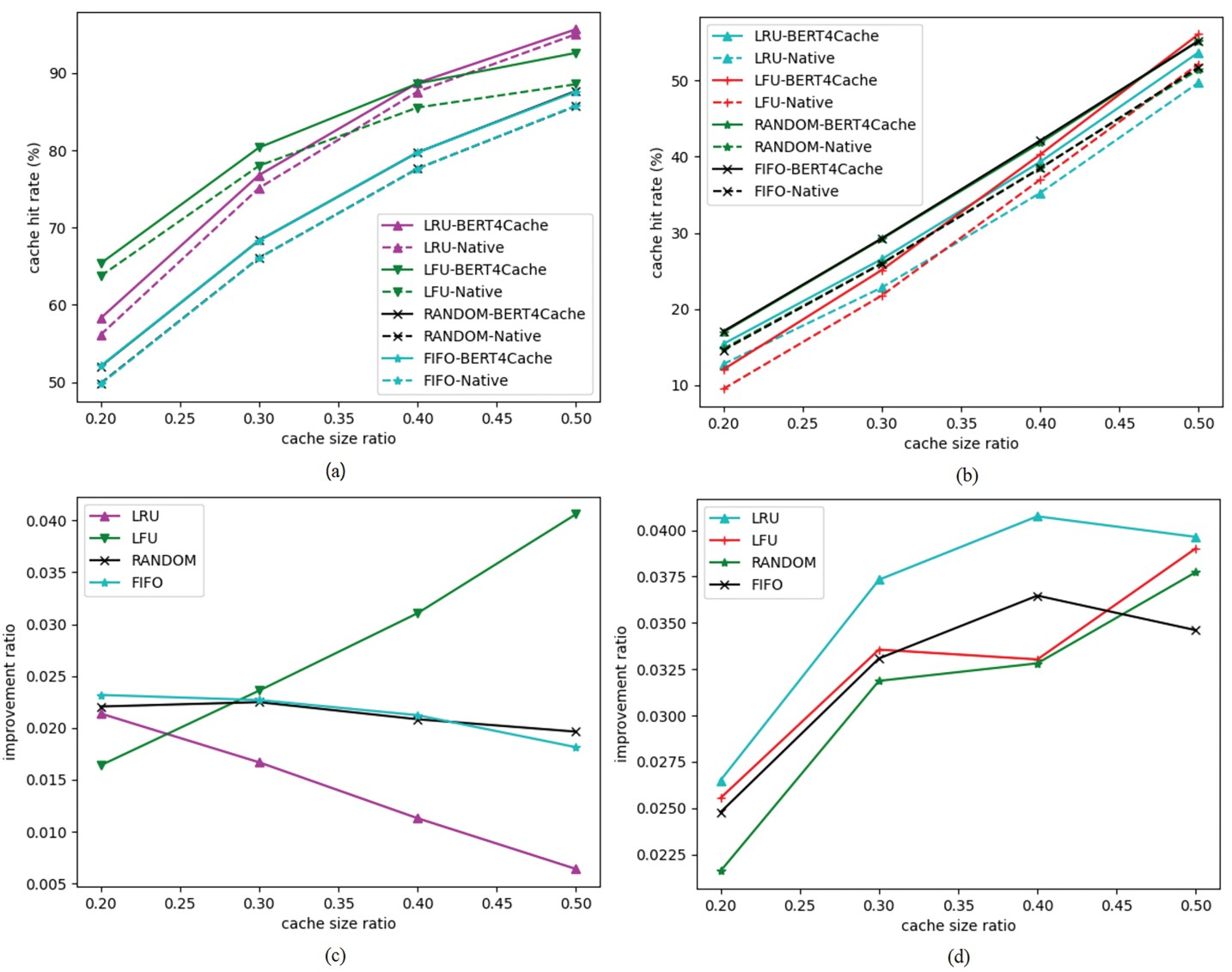

**Figure 3 Cache hit rate for high-frequency access objects.** Cache hit rates for high-frequency and low-frequency access data objects.

This finding highlights that the benefits of BERT4Cache are more significant in scenarios where cache space is limited.

### Performance comparison between high-frequency and low-frequency accessed data objects

High-frequency access data is accessed frequently by all users, while low-frequency access data is accessed infrequently by all users. We separately calculate the cache hit rates for high-frequency access data and low-frequency access data, comparing the improvement effects of BERT4Cache on the hit rates of these two types of data objects. Since BERT4Cache learns more information about the patterns of high-frequency access data,

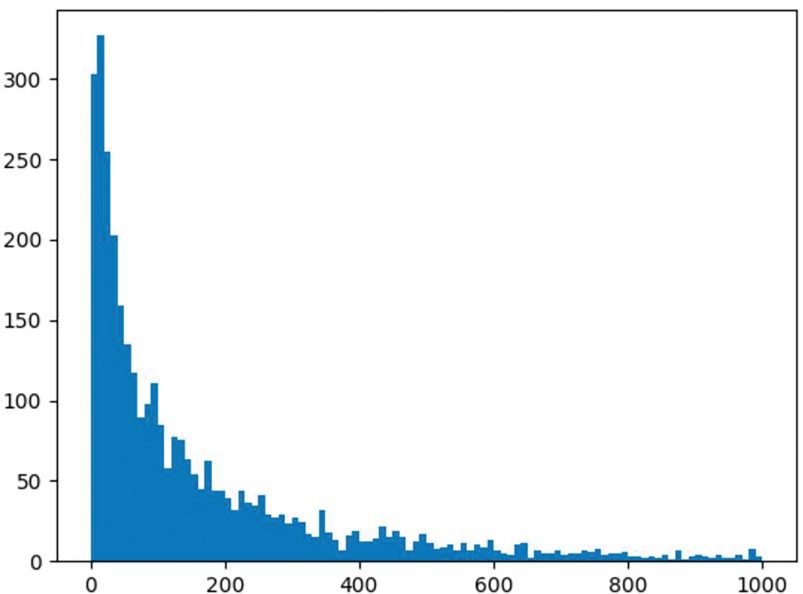

**Figure 4 Distribution of data object access frequencies.** This figure shows the distribution of access frequencies for all data objects, indicating that the majority of data objects have access frequencies of less than 200. We divide the data objects based on an access frequency threshold of 200, classifying data with access frequencies below 200 as low-frequency access data objects and those with frequencies above 200 as high-frequency access data objects, comparing their cache hit rate improvement effects at different cache ratios.

the improvement in hit rate for high-frequency access data should be greater than that for low-frequency data.

Figure 3 presents the cache hit rates for high-frequency and low-frequency access data separately. Figure 3C depicts the improvement ratio in cache hit rates for high-frequency access data objects, while Fig. 3D shows the improvement ratio for low-frequency access data objects. It can be observed that the improvement in cache hit rate is greater for high-frequency data.

Figure 4 shows the distribution of access frequencies for all data objects, indicating that the majority of data objects have access frequencies of less than 200. We divide the data objects based on an access frequency threshold of 200, classifying data with access frequencies below 200 as low-frequency access data objects and those with frequencies above 200 as high-frequency access data objects, comparing their cache hit rate improvement effects at different cache ratios. The experimental results reveal that BERT4Cache achieves greater improvements for high-frequency data objects compared to low-frequency ones. This discrepancy can be attributed to the fact that BERT4Cache relies on learning access patterns to make predictions. However, the access patterns of low-frequency data may be more challenging to capture, resulting in limited improvement in terms of prediction accuracy.

### *Improvement of cache performance through prefetching*

The caching simulation system implemented in "Settings and Dataset" only utilizes BERT4Cache for cache prefetching when there is a cache hit. In this section, we consider a

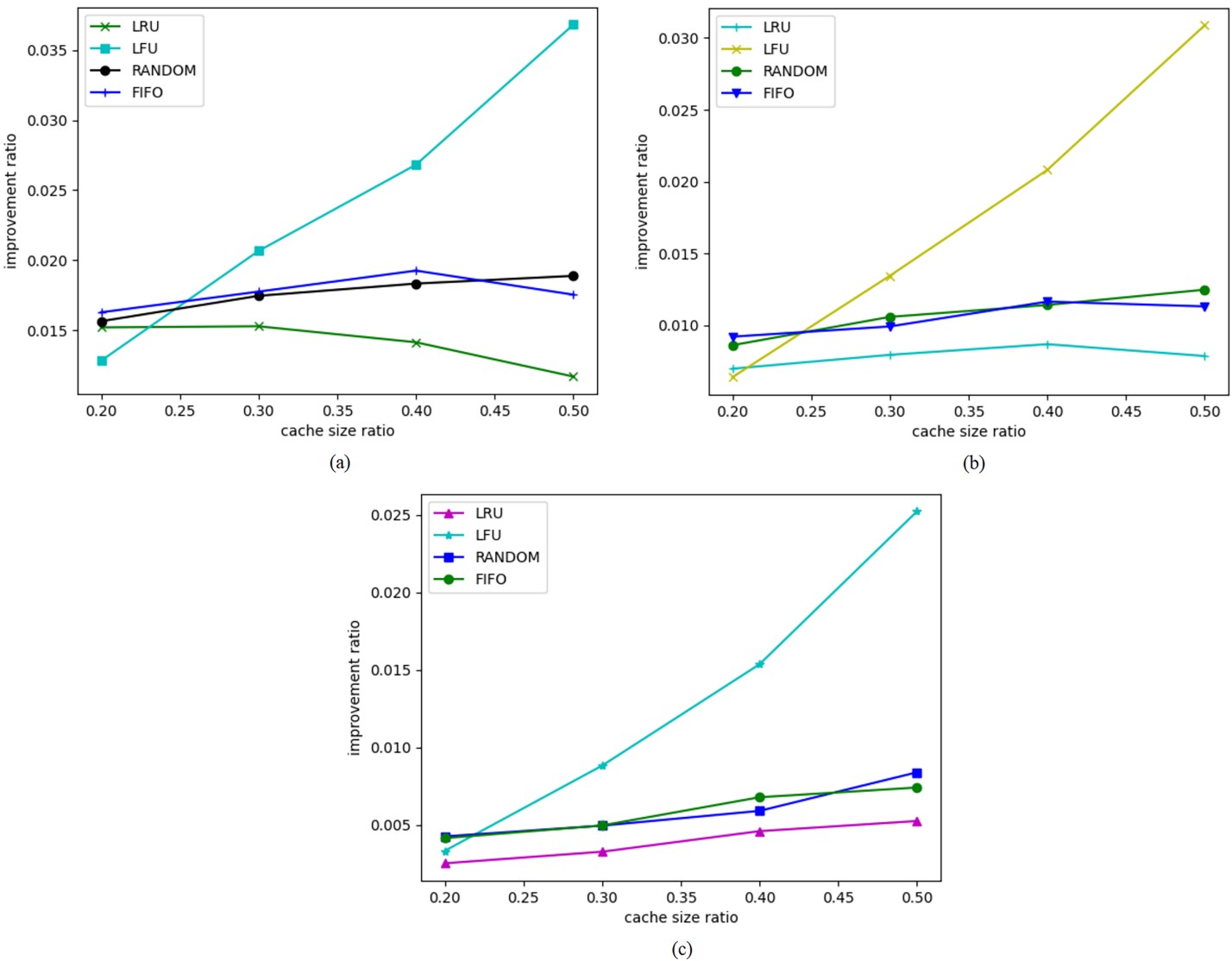

**Figure 5** $f_{ob}$ = **0.25.** Influence of different prefetch ratios ($f_{ob}$) on the improvement ratio of cache hit rates.

more aggressive cache prefetching strategy. When a user starts accessing the caching system, we wait for the user to access a certain number of data objects, denoted as $N_{ob}$. We then input this sequence of data objects to BERT4Cache to predict the same number of data objects, which are subsequently preloaded into the cache.

For different users, we set $N_{ob} = f_{ob}L_u$, where $f_{ob}$ is a parameter we define as the prefetch ratio, and $L_u$ is the length of the user access sequence. Figure 5 illustrates the improvement in cache hit rates for different values of $f_{ob}$. It can be observed that when $f_{ob}$ is set to 0.1, the greatest improvement in cache performance is achieved. When $f_{ob}$ is set to 0.25 and 0.4, too many data objects are preloaded, leading to a decrease in the accuracy of future access predictions by BERT4Cache and a reduction in the magnitude of the improvement in cache hit rates.

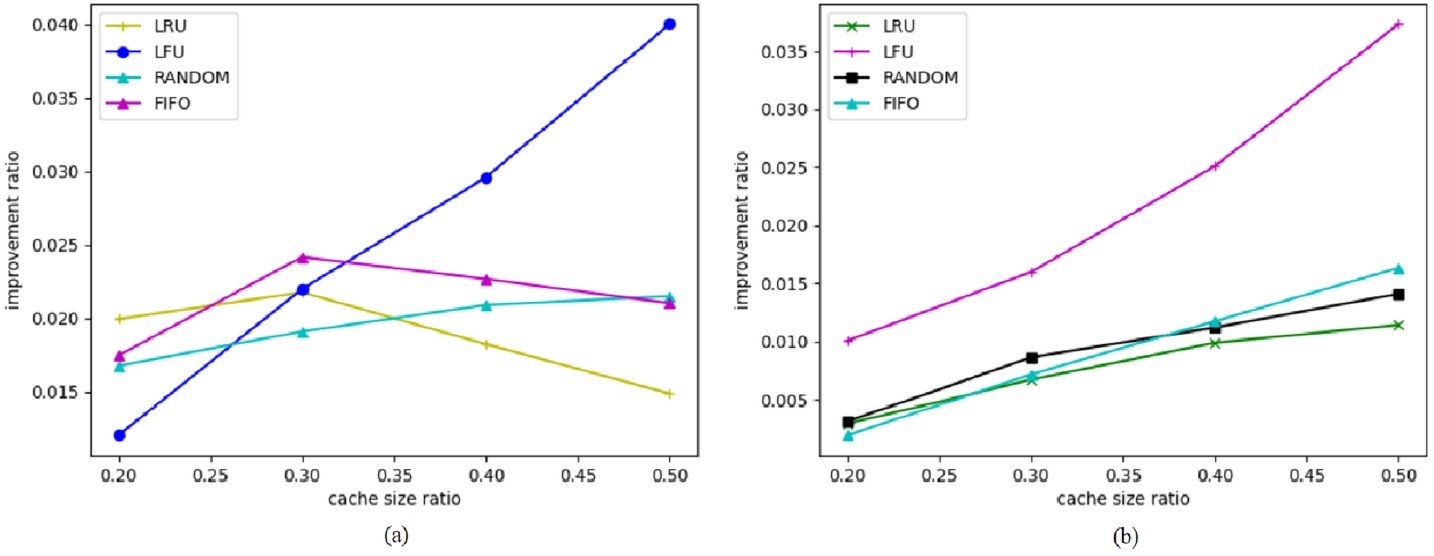

**Figure 6  Number of users: 10.** Improvement ratio of cache hit rates with multiple users accessing.

***Performance of BERT4Cache with multiple users simultaneously accessing***
In a real caching system, multiple users are served instead of just one, and each user has their own access pattern. We use BERT4Cache to improve cache hit rates in multi-user access scenarios.

For each user, we independently predict their access pattern and perform cache prefetching based on the prediction results. Figure 6 illustrates the improvement ratio of cache hit rates with BERT4Cache for different numbers of users. Compared to single-user access scenarios, the improvement diminishes, and the larger the number of users, the smaller the improvement.

## CONCLUSION

In this article, we propose BERT4Cache, an end-to-end model based on attention mechanisms and bidirectional Transformers. It learns users' sequential access patterns through cloze tasks and forward consistency tasks to predict imminent data requests, enabling proactive caching of predicted data objects before actual user requests are made. We comprehensively evaluate the generalization and robustness of BERT4Cache on publicly available datasets. The results indicate that BERT4Cache outperforms generic reactive and advanced proactive caching strategies. In comparison with LRU, LFU and FIFO, BERT4Cache achieves a 8% performance improvement. Compared to the latest methods ACORN and ANN-based proactive caching strategies, BERT4Cache achieves a 4.23% higher cache hits count than ACORN and a 1.5% higher cache hit rate than the ANN-based method. BERT4Cache has limitations, such as potential inaccuracies when user access patterns deviate significantly from learned historical patterns or when lacking relevant contextual information. Future work can explore strategies to mitigate these limitations and further improve BERT4Cache's performance.

### Funding

This work was supported by the National key R&D Program of China under Grant No. 2023YFB4503100, National Natural Science Foundation of China under Grant U23B2027, China Mobile Strategic R&D Project under Grant R23100LX. The funders had no role in study design, data collection and analysis, decision to publish, or preparation of the manuscript.

### Grant Disclosures

The following grant information was disclosed by the authors:
National key R&D Program of China: 2023YFB4503100.
National Natural Science Foundation of China: U23B2027.
China Mobile Strategic R&D Project: R23100LX.

### Competing Interests

All the authors are employed by China Mobile Information Technology Center.

### Author Contributions

- Jing Shang analyzed the data, performed the computation work, authored or reviewed drafts of the article, and approved the final draft.
- Zhihui Wu conceived and designed the experiments, prepared figures and/or tables, and approved the final draft.
- Zhiwen Xiao conceived and designed the experiments, performed the computation work, authored or reviewed drafts of the article, and approved the final draft.
- Yifei Zhang performed the experiments, prepared figures and/or tables, and approved the final draft.
- Jibin Wang analyzed the data, authored or reviewed drafts of the article, and approved the final draft.

### Data Availability

Data and code are available at Figshare:

Xiao, Zhiwen (2024). BERT4_Cache. figshare. Dataset. https://doi.org/10.6084/m9.figshare.25272634.v1.

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
