# Peer review of "BERT4Cache: a bidirectional encoder representations for data prefetching in cache"

_PeerJ Computer Science, doi:10.7717/peerj-cs.2258_

## Round 0.1 · original submission · Major Revisions

The reviewers have provided detailed comments regarding the technical and novelty, the author should prepare a major revision to improve the quality of their paper and make the contributions more clear.

Reviewer 1 ·

Basic reporting

The paper is generally in well-written, and easy to follow. The references, figures and tables are clear and useful.

Experimental design

This paper models user access behavior into Forward Consistency Pairs (FCP) and Sequence-Prefetch Sets (SPS) structures, and constructs a bidirectional Transformer model, learning the latent access patterns and preferences of data sequences, and applying them to cache prefetching. The effectiveness of the algorithm is verified through experiments. The work presented in the paper is innovative to a certain extent.

Validity of the findings

The findings presented in the paper includes modeling of sequential data, designing and training a BERT-like deep neural network, constructing a simulated cache system, and comparing it with classic baseline cache strategies such as LRU, LFU, and FIFO using publicly available MovieLens dataset, demonstrating its ability to simulate real user access patterns effectively. The performance of the proposed method is evaluated from multiple perspectives.

Additional comments

1. The current architecture diagram lacks necessary legend explanations, making it difficult for readers to understand the specific meanings of various module boxes, connecting lines, and other elements. It is recommended to add a legend at appropriate locations in the figure, briefly explaining the meanings of key symbols and colors represented.
2. The prefetching task shares certain similarities with recommendation systems, sequence prediction, and other tasks, as they all require modeling and predicting user behavior sequences. However, the authors chose a bidirectional Transformer architecture similar to BERT, rather than a traditional unidirectional autoregressive structure (such as LSTM). What were the main considerations behind this decision?
3. Why not model the entire cache access history sequence directly, instead of dividing it into multiple user sub-sequences? What are the motivations and considerations behind this partitioning?
4. Besides users, have you considered other ways of partitioning sub-sequences? For example, based on time windows, access sources, etc. How did you decide which approach to prioritize?
5. User sub-sequences may vary in length. Would the model be affected when dealing with variable-length sequences? Have you considered adopting strategies such as sequence splitting or truncation?

Cite this review as

Reviewer 2 ·

Basic reporting

As a reviewer, I have assessed the manuscript "BERT4Cache: A bidirectional encoder representations for data prefetching in cache" and offer the following ten points of feedback and suggestions for minor revisions, focusing on the method's innovation, the sufficiency of experiments, and the overall contribution:

The manuscript presents BERT4Cache as an innovative method for data prefetching in cache systems. The authors should clearly articulate the novel aspects of their approach and how it advances the state of the art in cache management.

The integration of bidirectional Transformers and attention mechanisms is a key aspect of the proposed method. The authors should provide a more in-depth discussion on the technical aspects that make this approach effective for prefetching tasks.
The experiments conducted demonstrate the effectiveness of BERT4Cache. However, additional experiments with different datasets or under varied conditions could further validate the robustness of the method.The paper compares BERT4Cache with generic reactive and proactive caching strategies. It would be beneficial to include comparisons with other contemporary methods to highlight the advantages more clearly.
The manuscript primarily uses hit rates as a performance metric. Expanding the set of metrics to include latency, throughput, and memory efficiency would provide a more comprehensive evaluation of the system's performance.
The MovieLens dataset is widely used, but the paper could provide more details on its preprocessing and relevance to the prefetching problem, ensuring that the dataset justifies the experimental setup.
The paper could benefit from a discussion on the interpretability of the model's predictions. Understanding how BERT4Cache makes prefetching decisions could be valuable for both users and system administrators.
The manuscript should address the scalability of BERT4Cache, particularly when dealing with large-scale systems or datasets, and discuss any potential scalability issues.
The paper lacks a detailed error analysis. The authors should include a discussion of the types of errors encountered and potential reasons behind them to provide insights for future improvements.
The paper should clearly outline the overall contribution of BERT4Cache to the field of computer science, particularly in terms of cache management and data prefetching.

For minor revisions, the authors are encouraged to:

Polish the manuscript for clarity and conciseness, ensuring that the contributions are well-defined and accessible.
Provide additional experimental data or analysis to support the claims made in the paper.
Ensure that all figures, tables, and mathematical notations are clearly labeled, described, and referenced within the text.
Include a discussion on the practical implications and potential real-world applications of BERT4Cache.

Overall, the paper presents a promising approach to enhancing cache performance through data prefetching. With minor revisions and additional details, this work could make a meaningful contribution to the field.

Experimental design

good. By the way, please give the metrics you used in this paper.

Validity of the findings

good

Additional comments

None

Cite this review as

Reviewer 3 ·

Basic reporting

This article fluently articulates the use of BERT technology for user behavior representation and enhances cache performance by maximizing cache hits through a temporal recommendation model.
It thoroughly investigates the application of machine learning techniques to address cache hits and innovatively applies advanced NLP technology to cache hit strategies.

The overall architecture diagram is somewhat obscure; it is recommended to first express the cache hit task and process, then clarify which stage's performance is primarily enhanced by BERT technology.

The formulas within the article should be centered for easier readability.

Experimental design

The experimental section of this article provides ample experiments to demonstrate the significant enhancement of cache hits by BERT. The experimental details are comprehensive.
It is suggested that the section clearly describe the comparative datasets and the relevant model descriptions.
The article can describe more analysis on why certain results were obtained for each parameter tuning experiment, rather than merely graphically presenting the proposed solution's superior performance on a particular metric.

Validity of the findings

The novelty of this article is commendable, applying advanced pretrained models to the field of cache hits and significantly improving cache hit rates.

It offers a new perspective on how to enhance cache hit methods for future research.

Cite this review as

Reviewer 4 ·

Basic reporting

In this paper, the authors provide BERT4Cache, a bidirectional Transformer model for data prefetch in cache. This paper highlights the problem of accurately predicting user interests and preferences, which is crucial for improving cache hit rates. It also mentions the use of BERT4Cache to predict imminent future requested objects and prefetch them into the cache. Experiments show that BERT4Cache achieves superior results compared to existing caching strategies. Overall, the paper presents a novel and effective approach for improving prefetch performance using a bidirectional Transformer model. The methodology is well-explained, and the experiments provide compelling evidence of the model's effectiveness. The paper is in well-written and easy to follow. I encourage the authors review more state-of-the-art methods published in recent two years.

Experimental design

The topic is interesting and falls into the readership of this journal. The research question is well-defined, and the experiment is solid. However, the key question that, why BERT is naturally suitable for prefetch in cache, should be introduced. If the authors can address this, I am sure it will receive more readership.

Validity of the findings

1) More detailed discussion should be provided on the limitations and potential challenges of BERT4Cache compared to other caching strategies.
2) The author should also demonstrate the robustness and generalizability of the model across different types of datasets and cache systems by discussing their technical differences.
3) Further analysis of the impact of hyperparameters and model parameters on the performance of BERT4Cache is also suggested.
4) The authors are also suggested to provide some cases study to show the superior advantages of using BERT4Cache.

Cite this review as

---

## Round 0.2 · accepted · Accept

The reviewers now recommend accepting this paper.

Reviewer 1 ·

Basic reporting

The paper is generally in well-written, and easy to follow. The references, figures and tables are clear and useful. The reviewers have shown a great advance in this revision.

Experimental design

This paper models user access behavior into Forward Consistency Pairs (FCP) and Sequence-Prefetch Sets (SPS) structures, and constructs a bidirectional Transformer model, learning the latent access patterns and preferences of data sequences, and applying them to cache prefetching. The effectiveness of the algorithm is

Validity of the findings

The findings presented in the paper includes modeling of sequential data, designing and training a BERT-like deep neural network, constructing a simulated cache system, and comparing it with classic baseline cache strategies such as LRU, LFU, and FIFO using publicly available MovieLens dataset, demonstrating its ability to simulate real user access patterns effectively. The performance of the proposed method is evaluated from multiple perspectives.

Additional comments

In the previous round of review, I have listed the following comments:
1. The current architecture diagram lacks necessary legend explanations, making it difficult for readers to understand the specific meanings of various module boxes, connecting lines, and other elements. It is recommended to add a legend at appropriate locations in the figure, briefly explaining the meanings of key symbols and colors represented.
2. The prefetching task shares certain similarities with recommendation systems, sequence prediction, and other tasks, as they all require modeling and predicting user behavior sequences. However, the authors chose a bidirectional Transformer architecture similar to BERT, rather than a traditional unidirectional autoregressive structure (such as LSTM). What were the main considerations behind this decision?
3. Why not model the entire cache access history sequence directly, instead of dividing it into multiple user sub-sequences? What are the motivations and considerations behind this partitioning?
4. Besides users, have you considered other ways of partitioning sub-sequences? For example, based on time windows, access sources, etc. How did you decide which approach to prioritize?
5. User sub-sequences may vary in length. Would the model be affected when dealing with variable-length sequences? Have you considered adopting strategies such as sequence splitting or truncation?
* * *
In this revision, the authors have addressed all of them well, therefore I have no further comments.

Cite this review as

Reviewer 3 ·

Basic reporting

This article is fluent and clearly explains how to use BERT technology to represent user behavior and then use the time series recommendation model to maximize cache hits and improve cache performance. The author provides a more detailed and intuitive architecture diagram with clear processes and innovations.

Experimental design

The authors supplemented detailed experimental data and model comparisons, as well as detailed algorithm comparison settings.

Validity of the findings

The novelty of this article is commendable, applying advanced pretrained models to the field of cache hits and significantly improving cache hit rates. It offers a new perspective on how to enhance cache hit methods for future research.

Cite this review as